# From Digital Collection to Open Access: A Preliminary Study on the Use of Digital Models of Local Culture

Chia-Ling Chang [1], Chin-Lon Lin [2,*], Chi-Hsien Hsu [3,*] and Yikang Sun [4]

1 Department of Education Industry and Digital Media, National Taitung University, Taitung City 950, Taiwan
2 Department of Cultural Creative Industries, Hungkuang University, Taichung City 1433, Taiwan
3 Department of Creative Product Design, National Taitung Junior College, Taitung City 950, Taiwan
4 College of Art and Design, Nanjing Forestry University, Nanjing 210037, China
* Correspondence: linpcl@sunrise.hk.edu.tw (C.-L.L.); chhsu@ntc.edu.tw (C.-H.H.)

**Abstract:** In the past, most cultural content was in a passive state of protection. In recent years, with the popularity of digital printing and the emergence of the concept of open-source sharing, it provides a new idea for cultural preservation. Using cultural elements from the Taitung region as a sample, this study established an open-source database, and completed the production of 60 digital models and the archiving of related materials. Based on the concept of open-source sharing, our research hopes that this database can be applied in more places. Through surveys, it could be concluded that, when the models are designed in parts and are easy to print and display, it is more conductive for the models to be used in promotions and applications. It is expected that each township will have its own localized 3D model database. Through the open-source localized digital model's unrestricted and free features, under the influence of COVID-19, it can also allow people from all over the world who cannot visit these places in person to print the local cultural content remotely, so as to have a three-dimensional under-standing of Taiwanese cultural objects. It is expected that the localized 3D model databases will help promote local cultural improvement and move towards local innovation.

**Keywords:** open-source; local culture; digital models; 3D printing

## 1. Introduction

In response to the rise of digital tools and open-source sharing and co-creation, digital manufacturing and open-source sharing have become new trends in the industry in recent years. Cultural content and objects should keep pace with the times and become closer to people's lives. Nowadays, digital manufacturing tools have become digitalized, affordable and popular. Open-source databases of 3D models for free downloading and printing are flourishing, and it is increasingly common for users to download what they print. The world's first 3D model open-source database, Thingiverse, emerged in 2008. In Taiwan, the first 3D model search platform, Yobi3D, was established by a non-governmental organization in 2014, and in the following year, the Industrial Development Bureau of the Ministry of Economic Affairs also established an official 3D model platform, Fast Lab. To date, there are dozens of common 3D model open-source platforms domestically and internationally. However, the content of the platforms is only about dolls, mechanical parts, and household items. The applications of 3D printing worldwide are primarily on consumer products (29%), automobiles (19%), medical care (13%), education (10%), space (8%), and industrial machinery (7%). Moreover, 3D printing technology has been used in various fields, such as society, technology, education, and medicine, with varied performances. Tseng and Wang (2019) applied 3D printing to prosthetic limbs through teamwork and open-source design, making the process faster and more relevant to users [1]. With the emergence of 3D printers that can print circuit boards or food, the content subject to 3D printing has been much more than ever before, but the cultural content is rarely a digital print target.

In the past, cultural contents have been passively preserved in the form of the original artifacts, texts, images, audio and videos, and guiding websites, with the goal of long-term collection and appropriate preservation. In view of the maturity of 3D modeling technology, this study proposed the concept of open-source local culture, in which local cultural contents are assembled into a digital database platform in the form of 3D modeling. The platform introduces the spirit of open-source sharing and popularized 3D printing technology, which will enable more people to actively download local cultural contents remotely and print them for value-added applications. With the technological changes of the times, cultural relics have been preserved in various ways and the purposes of presentation vary at different times. From 2002 to 2007, the National Science and Technology Council conducted a digital collection to select culturally valuable objects, with the main purpose of preserving cultural objects in the form of photos, videos, and illustrations. Since 2008, the Ministry of Culture has been promoting the cultural creative industry by adding creative value to cultural contents through various translations in 13 other categories, such as commodity packaging, cultural and creative goods, film and television, music, and animation and comics. In response to the changing times and industrial model, the government and industry have encouraged the connection of key texts such as words, images, and stories, and the diverse licensing of intellectual property (IP) has become the key to unlocking cross-disciplinary products. Liao and Chen [2] pointed out that the integration of digital content, user services, and technological tools through technology in the local culture is conducive to the intersection of culture and technology. For example, in the publishing industry, through IP licensing, creative texts are extended to film and television, games, comics, music, and performances. In light of digital tools and the concept of open-source sharing, cultural contents should keep up with the times and be more relevant to people's lives. Under the trend of globalization, "localization" is exactly a valuable asset for Taiwan to express its identity to the world. The booming digital technology and open-source networks are good ways for the world to see Taiwan's local culture [3]. The creation of open innovation platforms by governments or citizens, as a powerful bridge to user innovation, can also be seen as a social movement [4].

The purpose of this study is to build an open-source database of cultural content in Taitung, which is a disadvantaged area in terms of scientific and technological resources, and to explore the print application and impact of the open source of local cultural content as a reference for the subsequent open-source process of local cultural contents. Taitung is rich in natural products and cultural diversity; however, due to its remote geographical location, digital resources are comparatively scarce out there. From the view of fulfilling the social responsibility of universities, universities should have gratitude for the source of benefit. It is expected that more people can get close to Taitung's local culture through digital innovation, thus highlighting the potential of digital design in remote areas to transform knowledge circulation under the trend of open-source sharing.

## 2. Literature Review

### 2.1. Content and Connotation of Local Culture

Culture is the soul of a nation, and any culture with historical, artistic, and scientific values for cultural preservation is a precious national asset. From the perspective of locality, local culture is essentially a reflection of people's local life and culture. It is by making good use of local cultural assets, such as geographic resources, human history, and folklore activities that the unique charm of a place can be built. As globalization is progressing all around us, the exchanges between different cultures are becoming more frequent, and cultural globalization seems to have served as a driver of "cultural convergence". In fact, while globalization is progressing, the trend of respecting multiculturalism and emphasizing local culture has never diminished. Lee [5] argued that culture is the product of common human activities, which includes not only the tools and objects that people use, the rules and regulations that sustain social life, and the artistic products of spiritual life, but activities of the human mind in the process of creation [6,7]. Each place has its

own local symbol, and a "local place" is a local, regional, and localized object, location, field, or area which sets itself apart from anywhere else in the world by its local features, such as atmosphere, characteristics, and style. The content of cultural assets can be divided into tangible and intangible (material and non-material), which are the result of cultural accumulation in the past and leave historical traces in the places where people live today, some of which are tangible structures or objects, while others are non-material skills and arts [8].

In the Cultural Heritage Preservation Act, states: "The term "cultural heritage" referred to in this Act means the following designated or registered tangible or intangible cultural heritages which are of cultural value from the point of view of history, art or science. Tangible cultural assets include nine categories of monuments, historic buildings, monumental buildings, groups of buildings, archaeological sites, historic sites, cultural landscapes, antiquities, natural landscapes, and natural monuments; and intangible cultural assets include five categories of traditional performing arts, traditional craftsmanship, oral traditions and expressions, folklore, and traditional knowledge and practices".

At the local level, it is indeed necessary to have a cultural core to lead the development of local characteristics. Regarding the classification of local cultural assets, many scholars have followed Professor Kiyoshi Miyazaki's five categories of "people, culture, place, production, and landscape" to classify local cultural resources [9,10]. The core content of these five points is as follows: (1) People refers to the satisfaction of the common needs of local residents, the management of interpersonal relationships, and the creation of well-being in life. (2) Culture refers to the continuation of local history and culture, the management of artistic and cultural activities, and lifelong learning. (3) Place refers to the maintenance and development of the geographical characteristics of a place, and the emphasis on local characteristics. (4) Production refers to the development and marketing of local products, and the collective promotion of local economic activities. (5) Landscape refers to the creation of unique local landscapes, the sustainable management of the living environment, and the self-involvement of the residents in the development of the local area.

In Taiwan, the classification of local culture includes local tourism, craftsmanship, maintenance of cultural assets, and preservation of groups of monuments, such as the creation and publication of fine arts, and the establishment of culture and arts, living arts, agriculture, and fishery [11]. Local culture is essentially a manifestation of local life and culture to build up the unique charm of the place based on local thinking, using local resources, talents, and conditions, which is an endogenous and autonomous re-creation in the area [12]. Local culture is marked by the characteristics of local culture, whether from the five categories of people, culture, place, production, and landscape, the four categories of life, production, ecology, and living creatures, or the three categories of culture & history, technology, and nature, all of which are the contents of cultural industry [8]. In terms of practical cultural design, cultural content can be divided into three levels: physical or material, social or behavioral, and folk or religious [13]. In addition to the academic perspective, in the practical design of the industry, Cheng [14], with years of experience in the design of local culture and creativity, summarized the content of local culture in his book into local culture and history, ecology, nature, legend, and craft. This study synthesizes the above academic theories and practical views and proposes the orientation and details of a local cultural content survey, as shown in Table 1.

## 2.2. Open-Source Sharing Age

The term "open source" emerged in 1998. In a modern society where knowledge is exploding and resource sharing is a necessity, an open-source attitude means transparent sharing and cooperation with the public. The term "open source" originally referred to a mechanism that opens up its design for free modification by all users and was mostly used in the software development process, nut nowadays this mechanism has gradually evolved into a concept and even an attitude in life. It means the way of accelerating the development of products, projects, and programs by opening them up to public participation, discussion,

and modification, thus increasing transparency and public welfare [15]. Hacker culture and the free software movement are two examples of open source; they are seen as part of the progress even when they failed [16].

**Table 1.** Orientation of local cultural content survey.

| Literary and Historical Survey | Ecological Survey | Landscape Survey | Craft Survey | Folklore Survey |
|---|---|---|---|---|
| Historic buildings | Agricultural specialties | Mountain | Traditional craftsmanship | Religious stories |
| Temples | Native/Specialty plants | Rivers/Seas | Totems/Signs | Festivals |
| Featured buildings | Native/Featured animals | Flatland | Technical talent | Folklore stories |
| Cultural landscape | | Tourist attractions | | Legendary stories |

Source: this study.

Open-source sharing offers a fourth mode of exchange in the world which is not under the premise of profit. The current world of information has entered an era where scarcity, distribution, and hierarchical modes of exchange are unsustainable. At this time, the open-source community provides a model that begins with "sharing to non-specific people" and proceeds through the process of "gathering contributions from all" to enable sustainable, successive, and shared creations to occur. In the public press release of the COSCUP'14 held at Academia Sinica in 2014, it was clearly pointed out that open source is not only about software but also about a spiritual attitude. Using the spirit of openness, COSCUP aims to make its technology available to others without royalties and to evolve into a model of collaborative development, which is widely used in computer science. The nature of open source can be likened to "a single spark can start a prairie fire". People can play and contribute their own power to contribute to society with a good idea and a good platform [17]. Because of its open nature and ability to quickly gather the power of the masses, the spirit of open source can often do, in a very short time, what one person cannot. Open source is often a distributed and decentralized approach to collaborative development, and the open licensing of research and creative output is a prime example of user innovation [4].

Today's information society owes its existence to the profound influence of the free and open-source culture. Furthermore, the current global eagerness in promoting 3D printing technology is also from the development of an open-source technology project called Reprap by Adrian Bowyer of University of Bath (UK). The research and development process is all based on an open-source file (open-source technology) of hardware and software, and all the relevant technical data are publicly available on the Internet. As a result, this product, which should have stayed in super labs or have been belonged to high-end technology for large-scale economic enterprises, has found its way into the life of the general public in just a few years and is increasingly popular among individuals and families. Nowadays, the applications of increasingly sophisticated online trading platforms are becoming more and more diversified. With the wide and far-reaching super-influence of the online community movement, when everyone can own a desktop digital manufacturing tool (e.g., 3D printers and laser cutters), industry and society are bound to embrace the new technology. EI Bedewy, Lavicza, Haas, and Lieban (2022) provided teachers and students with the open-source tool "GeoGebra" for learning the connection between math and architectural modeling, and the results of this study indicated that participants were able to solve the technical problems of model visualization more easily through open-source modeling resources and 3D printing processes, thus effectively enhancing the learning opportunities in STEAM education [18].

Taiwan has also been influenced by the spirit of open-source sharing, mainly by software workers in the early days. However, with the development of the licensing and self-creation culture of CC in Taiwan, people who work on texts, hardware, music, videos,

designs, education, and politics are now also participating in open culture [19], and the scope of resources is becoming more and more extensive. The Taiwan CC Project uses Mediagoblin as a benchmark for open-source licensing to release CC or materials in the public domain [20]. Open sources also include other text (such as Wikipedia), hardware (such as Arduino and Thingiverse), music (such as Blend and SoundCloud), video (such as YouTube and Flickr), design (such as Behance), education (such as Khan Academy and OCW), science (such as arXiv), and politics (such as g0v). Taiwan's first Digital Minister Audrey Tang is an open-source leader and one of the few people in Taiwan with international open-source community influence. In terms of cross-discipline, the popularity of input tools (touch, voice, gesture, and emotional signals) has brought more and more analog messages into the digital world, and the corresponding collaborative space has continued to decline its operational barriers and, along the new output methods (stereoscopic printing, augmented reality, and programmable matter), came into life. The content of open source is much more inclusive than a complex of print, audio, and text media.

In view of this, the issue of how to open up and provide multiple applications for the digitization of cultural contents has become an issue of concern. According to the Center for Digital Culture at Academia Sinica, Taiwan, the dissemination and circulation of knowledge in the digital age are based on the digitization of cultural content and the continuous evolution of information technology. In today's world where "knowledge has a price", how to enable the general public to appropriately use open and authorized data in a free network environment, and thus promote the co-creation and progress of local culture, has been the subject of continuous discussion and practice by various collection institutions. The former convener of the Center for Digital Culture, pointed out in the Open Museum [21] that "opening up" the right to use the collections, which includes the collection (determining the value of the artifact), interpretation (interpreting the meaning of the artifact) and creation (using the artifact to create), is open to all people and promotes the "democratization" and "digital affirmation" of cultural assets, thus facilitating the positive cycle of knowledge production through the multi-directional and multi-dimensional digital display. Cheng [22] proposed three features of the Open Museum. One feature is that the digital images of the collections are in the International Image Interoperability Framework (IIIF) format, which allows direct online viewing of large images regardless of the carrier. The second feature is that the collections have a higher chance of being found by search engines. The third feature is that the Open Museum's Online Exhibition Module service connects the collections with the exhibitions, allowing the public to click directly from the same page without any obstacles. From the above features, it is clear that the process and services of cultural content from collection to openness are all based on the idea of "easy access for the public".

### 2.3. Prototyping Technology and Self-Maker Spirit

Rapid Prototyping and 3D printing technologies have been developed since the 1980s. Today's 3D prototyping technology is different, but the main printing process is similar. First, 3D modeling is carried out by computer-aided design software, and then the finished design file is sliced by slicing software, and each slice contains the inner and outer contours of the product. Finally, these slice profiles are converted into G-code parameters to control the printing values of 3D printers [23,24]. Over the years, the process has continued to break new ground. The open-source community has played an important role in this wave of development because since 3D printing hardware and software have been made available to the community, talented engineers from all over the world have been able to collectively work to achieve the ideal of making 3D printing technology available to the general public. This community environment has led to a wide variety of new printer models, and the general public can easily evaluate the print quality, print speed, printable size, output stability, and price of various models, so that everyone can achieve the dream of "freedom to print and make any object" by choosing a 3D printer that meets their needs. With the help of open-source hardware and the Internet community, due to the advancement of

technology, the wave of self-makers has risen, and product manufacturing has moved from factories to homes and personal studios. The influence of digital software resources and smart machines on manufacturing has created a new wave of social and technological revolution [25], as digital manufacturing tools have become affordable and accessible, changing the way society works.

The digitization and personalization of manufacturing tools are not only for researchers or creators, but more importantly, they greatly affect individual self-makers and general public users. Each person can design according to his or her free creativity and functional needs and make the work that best meets his or her expectations [26]. Furthermore, through the Internet, it will be easier for each self-maker to open up his or her design information and manufacturing knowledge to the four corners of the world and share it, further breaking down social and cultural barriers. Lipson and Kurman [25] proposed that self-makers use digital manufacturing tools and various processing methods to connect, either directly or indirectly, digital data with various materials such as paper, wood, fabric, resin, and metal. While exploring the relationship between materials and processing methods through computers, and exploring the multiplication effect between the two, they also create data and objects. This is a different kind of production method from the traditional one, in which digital manufacturing tools connect abstract digital information with concrete materials. While they are the extremes of each other, a two-way communication is established between them. Moreover, 3D printers are inevitably becoming more affordable and personal, and the biggest source of profit from 3D printing in the future will not be the machines themselves or the consumables, but the image files and databases. With the development of 3D printing technology, the 3D printing industry has developed into a multi-billion-dollar market and continues to grow. Chang and Tsai [27] analyzed the features and community functions of several domestic and international digital modeling platform databases. In these databases, users are free to access high-quality, usable, and secure image files from model databases. Furthermore, enterprises and manufacturers can manage user-downloaded image files from databases and even have access to target users' biographical data.

Continuing the spirit of self-makers, with government subsidies and vigorous promotion and support from non-governmental organizations, Maker Spaces have blossomed in Taiwan. Since 2012, there have been many self-maker spaces in universities, colleges, senior high schools, vocational high schools, junior high schools, and elementary schools with excellent digital manufacturing tools, among which 3D printing equipment is the most common digital manufacturing tool. In principle, digital modeling combined with 3D printing is user-centered and both are considered as active and self-directed learning methods [28]. In school teaching, Bonorden (2022) used digital models of flowers in biology classes and 3D printed three-dimensional models to solve the problem of having only 2D pictures of plants and flowers in textbooks. The three-dimensional models clearly illustrated flower structures, thus improving the limited learning environment [29]. However, the prerequisite is that this can only be achieved if the teacher has modeling skills [30,31]. After a deeper understanding, this study finds that although there is no shortage of space or hard equipment, there is a lack of technical talents and digital creation content. Only a few professional teachers have modeling skills [32–34], and for most, it still takes a lot of energy and time to learn these new digital technologies [30,31]. The 3D open-source databases can solve this problem in a timely manner. Through CC authorization, they provide suitable digital 3D files for users to download and print freely to produce three-dimensional models and solve the dilemma of insufficient digital content and technology. Users can master what they learn. However, this can only be successfully practiced if the teacher has professional modeling skills, which is the prerequisite [30,31].

## 3. Research Method

Based on the concept of "cultural collection and re-creation" and to fulfill the social responsibility of universities, this study aims to investigate and inventory the local cul-

tural contents in Taitung. It selects the appropriate modeling themes and related objects through the principle of selection and records the size, material, and cultural and historical descriptions of the physical objects. The purpose of this project is to confirm the accuracy of the information on the objects created for knowledge dissemination and educational applications. The 3D dynamic model is constructed through staff training, and slicing software analysis and G-code are performed to make feasible the model printed. Finally, the 3D model files are uploaded to the open-source platform databases, the usability of the open-source platform is discussed, and the traffic data are analyzed to complete the open source of the local cultural contents in Taitung.

*3.1. Inventory and Survey of Local Culture*

There are three reasons why Taitung is selected as the target area. Firstly, Taitung is the third largest county in Taiwan by area, with a beautiful natural landscape, diverse human and cultural communities, and rich local cultural contents. Secondly, because of the far-reaching nature of Internet technology, Taitung, located in a remote rural area compared to other counties and cities, requires more aid to break the urban–rural boundary through the power of new digital technology. Finally, the researchers and their affiliated organization have the advantage of geographical location and connections in Taitung, which is conducive to this study.

Taitung County has one city, two towns, and 13 townships. Because of its late development, Taitung retains a rich aboriginal culture. According to the statistical report of the Taiwan Council of Indigenous Peoples Taitung's indigenous population accounts for more than 30% of Taitung's population, the highest in Taiwan. From the highest to the lowest indigenous population, the order is the Amis, Puyuma, Paiwan, Bunun, Rukai, and Atayal. In terms of tangible assets, aboriginal cultural relics and crafts of each ethnic group are quite diverse and rich. For example, the Puyuma are good at weaving rattan or bamboo utensils, using such techniques as square weaving, herringbone weaving, and hexagonal weaving to weave rattan baskets, rattan bags, and backpacks; the Paiwan have earthenware pots, glazed beads, bronze swords (three treasures of them), as well as cups and earthenware beads; their antiquities include human-animal-shaped jade penannulars and frog-shaped jade ornaments of the Hualgang Mount culture, whose totemic motifs all have deep cultural connotations of the aboriginal people. In terms of landscape and ecology, Taitung is bordered by the Pacific Ocean to the east, located in a tropical climate zone and facing the mountains near the sea, with a coastline of 176 km, the longest in Taiwan [24]. Taitung is also home to many animals, including wild boars, Reeves's muntjac, Rusa unicolors, and Pteromys volans in forests; Taitung also has many famous marine animals such as the dolphinfish brought by the tide, marlins in Chenggong Township, and Cypselurus in Pongso no Tao. In terms of architectural resources, Taitung has 47 featured buildings, such as Chinese Consolidated Benevolent Association, Taitung Thean Hou Temple, Baoting Art and Culture Center, Guanshan Station Kanzan, GoBen Farm, YiWan Taiwan Presbyterian Church, Lyudao Lighthouse, Lanyu Weather Station, and the chapel of St. Joseph Technical High School. In addition, the only Taitung architect in Taiwan e-Learning and Digital Archives Program of National Science and Technology Council—Mr. A-Yu Lu, whose architectural works, such as Taitung County Council, the Old Beinan Township Office, the Baosang Road Building, the Cave House (the old office building of Taitung County Tax Bureau), and San Hai Department Store, (shown in Figure 1) though an amateur, occupy an important place in Taitung's architectural history with his distinctive architectural style. In terms of intangible assets, Taitung's most well-known folklore belief of Bombing Lord Handan in the Lantern Festival enjoys equal popularity as Yanshui Beehive Fireworks Festival and Pingxi Lantern Festival for being the three major Lantern Festival folklore events in Taiwan. The indigenous people's festivals include the Paiwan Five Year Ceremony, Puyuma Mangayaw (Indigenous Hunting Festival), Tao Flying Fish Festival and Boat Launching Festival, and Bunun Ear Festival (Wikipedia: Taitung County Government, Taitung County

cultural assets, Taiwan's Indigenous Peoples Portal, and Taitung County of Indigenous People website). The relevant indigenous people's festivals are shown in Figure 2.

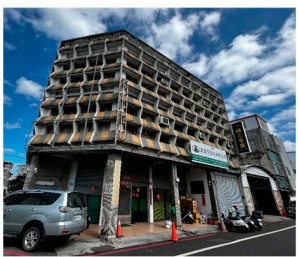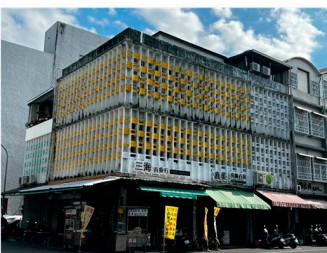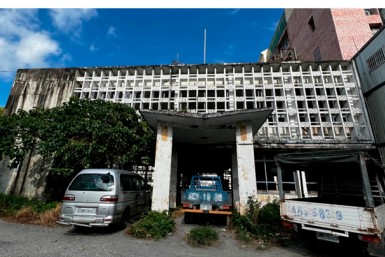

**Figure 1.** Diagram of Taitung's cultural assets. (Source: this study).

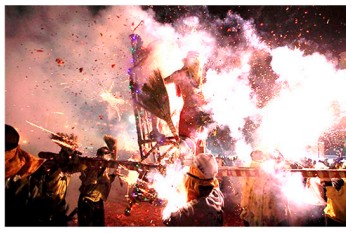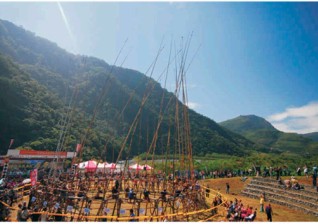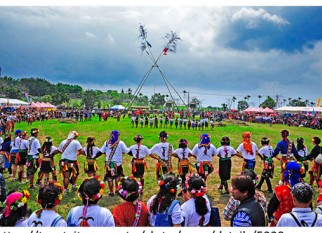

https://tour.taitung.gov.tw/content/images/static/3-6-12-0.jpg    https://tour.taitung.gov.tw/zh-tw/attraction/details/1272    https://tour.taitung.gov.tw/zh-tw/news/details/5229

**Figure 2.** Indigenous people's festivals (Source: website of Taitung County Tourism Department).

Taitung is rich in land resources and diverse in humanities. However, due to its remote location, it is comparatively lacking in technological resources, so it is more appropriate to adopt a "proximate view" to select modeling objects and consider the difficulty of 3D printing objects. In the open-source network world, rural areas can also be the protagonists in the field of digital modeling, and a unique "digital culture experience in Taitung" can be constructed.

Through the above collected Taitung cultural contents, the following three principles of modeling priorities were established: (1) uniqueness and representativeness of local and cultural significance: according to the classification of local cultural industries by Liao [8] and Cheng [11], and visiting local cultural communities Taitung Sustainable Development Society and Taitung County Houshan Association of Cultural Work for their suggestions on local characteristics of Taitung; (2) the urgency of preservation: for instance, historical buildings are vulnerable to natural disasters, (3) popularity and celebrity: attracting the audience attention and leading to traffic for downloadable applications.

In this study, the Taitung culture digital model was developed by integrating five major themes, as shown in the table below, and the specific modeling objects are described below: (1) cultural artifacts: mainly Taitung's aboriginal artifacts and Puyuma ruins; (2) local ecology: Taitung's common forest animals and fish; (3) featured plants: mainly Taitung's representative agricultural products; (4) featured buildings: Architect A-Yu Lu is an important figure in Taitung's architectural history. He is the only Taitung native in the Ministry of Science and Technology's Digital Archives Program for Taiwan's architectural history, and the most popular tourist attraction of the Lyudao Lighthouse; and (5) cultural festivals: the people, places, and props involved in Taitung's unique religious festival of "the Bombing of Master Handan", as shown in Table 2:

**Table 2.** Modeling themes and related objects in Taitung.

| Cultural Forms | Topics | Description | Related Modeling Objects |
|---|---|---|---|
| Tangible Assets | Cultural artifacts | Original folk crafts, totems, and decorations | Paiwan's bronze swords, glazed beads, earthenware pots, and cups. |
| | Local ecology | Terrestrial organisms, marine organisms | Wild boars, muntjac, Rusa unicolors, Cypselurus, marlins, and dolphinfish. |

**Table 2.** *Cont.*

| Cultural Forms | Topics | Description | Related Modeling Objects |
|---|---|---|---|
| | Featured Plants | Agricultural specialties | Sugar apple, Hibiscus sabdariffa, Navel oranges, a-bai, Areca catechu. |
| | Featured buildings | Featured buildings, historic sites | Taitung Country Council, the Old Beinan Township Office, Cave House, Kwong Hang Fat Building, and San Hai Department Store. (A-Yu Lu's buildings) |
| | | Famous landmarks | Lyudao Lighthouse. |
| Intangible Assets | Folklore | Folklore, religious activities | The Bombing of Master Handan (Personnel: fleshly Master Handan, palanquin bearer, cannon thrower; Location, Props: Xuan Wu Tang temple, shrine seat, firecrackers, banyan leaf fan, broom. |
| | Festivals | Aboriginal festivals | The number of characters, clothing features, and props required for the Five-Year Ceremony. |

Source: this study.

In addition to the above-mentioned modeling selection principles, this study also developed the principle of the print ratio, which was set at 1:100 for architectural printing, in response to the range of printing sizes of common 3D printers.

*3.2. Cataloging Object Data*

The metadata was set with reference to the digital collection databases and museums' collections and adjustments and revisions were made. The data catalog includes object names, 3D modeling drawings, and physical product print photos. There are four key points in the catalog: (1) Provide at least three keywords for each object, so that users can find it quickly, easily search cultural contents and related works, and increase an object's chance of being viewed; (2) Provide a "summary description" of an object, covering its actual size, the cultural connotation of the elements, and the design concept of the creator, to make it understandable for the public; (3) Provide digital model "3D printing parameters", such as Format, Printer, Rafts, Supports, Resolution, Infill for users' reference printing to reduce the chance of print failure and enhance the access value; (4) Adopt the six licensing terms of CC (creative commons), which have been promoted internationally in recent years. The object information cataloging format as shown in Table 3.

**Table 3.** Object information cataloging format.

| Object Name | | Object Code | |
|---|---|---|---|
| **1. Basic Description** | **2. Cultural and Historical Descriptions** | **3. Print Parameter Information** | **4. Pictures** |
| 1.1. Keywords<br>1.2. Size<br>1.3. Material<br>1.4. Color | 2.1. Cultural significance<br>2.2. Importance<br>2.3. Location | 3.1.1. Extrusion layer thickness<br>3.1.2. Extrusion width<br>3.1.3. Shell (surface) thickness<br>3.1.4. Number of outer circles | 4.1. Original view of the object<br>4.2. Object digital modeling diagram<br>4.3. Screenshot of printing parameters<br>4.4. Object print completion diagram |
| | | 3.2.1. Raft style<br>3.2.2. Support and object contact surface clearance setting value<br>3.2.3. Support angle values<br>3.2.4. Side skirt setting value | |
| 5. CC License Terms | | | |

Source: this study.

*3.3. Building 3D Digital Models*

The 3D models are built according to the modeling themes proposed above, and the modeling software is not limited to the creator's expertise and suitable attributes. Modeling

software in this study was mainly built with Maya, Z Brush, and Sketch up. Maya and Z Brush are suitable for building models of people, plants, animals, and artifacts, while Sketch up is suitable for building models of architecture. After the models were built, they were saved in .stl and .obj formats, which are two highly versatile 3D printing model file formats, and were used as the 3D digital model files for this study.

### 3.4. Confirm 3D Printing Parameters

From past 3D printing experiences, it can be learnt that not all 3D digital models can be printed smoothly, and this is the main reason why ordinary users have difficulties in 3D printing. For example, 3D models that are nearly 90 degrees or in the air may not always be able to be printed smoothly and must first be analyzed by slicing software to calculate the external support structure and help convert the 3D file into a slicing file, and then adjust the relevant parameters. This is to prevent the model from collapsing and failing when the user prints. Therefore, in this study, considering the ease of printing and convenience, each model was successfully printed and tested before uploading to the platform, and the values of the slicing parameters at the time of successful printing were also fully recorded to improve users' printing success rate, taking the Yin-Yang pottery pot as an example, as shown in Figure 3.

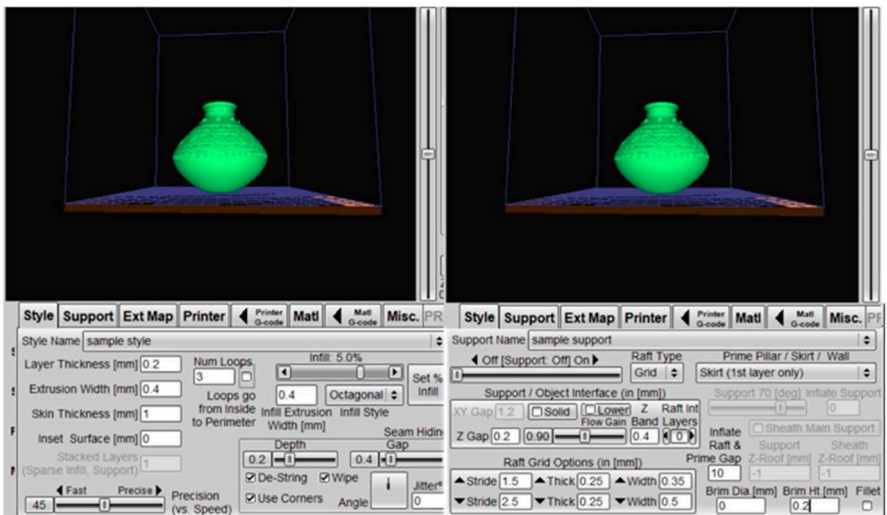

**Figure 3.** 3D printing of pottery pots with slicing parameter values. (Source: this study).

In 2020, this study published "Discussion on Common Problems and Solution Strategies in the 3D Printing Process", proposing 16 solutions to common problems in 3D printing. The 3D printing instruction focuses on basic parameter settings including the platform temperature, printhead temperature, printing speed, printing layer, outer wall thickness, density fill, base, and support material calculation. Before making a test print, it is important to check the scale of the model and make sure that the model does not violate the conditions and settings of the 3D printing device before printing.

### 3.5. Uploading Open-Source Platforms

A total of 60 digital models of Taitung culture were completed in this study, and the exhibits were presented on the platform as Figure 4. All of the works have been uploaded to the world's two most well-known and most-used open-source platforms, Myminifactory [https://www.myminifactory.com; (accessed on 20 January 2022)] and Thingiverse [https://www.thingiverse.com; (accessed on 20 January 2022)], and an account has been set in the Taitung Culture Content Zone for future file management, updating, and analysis.

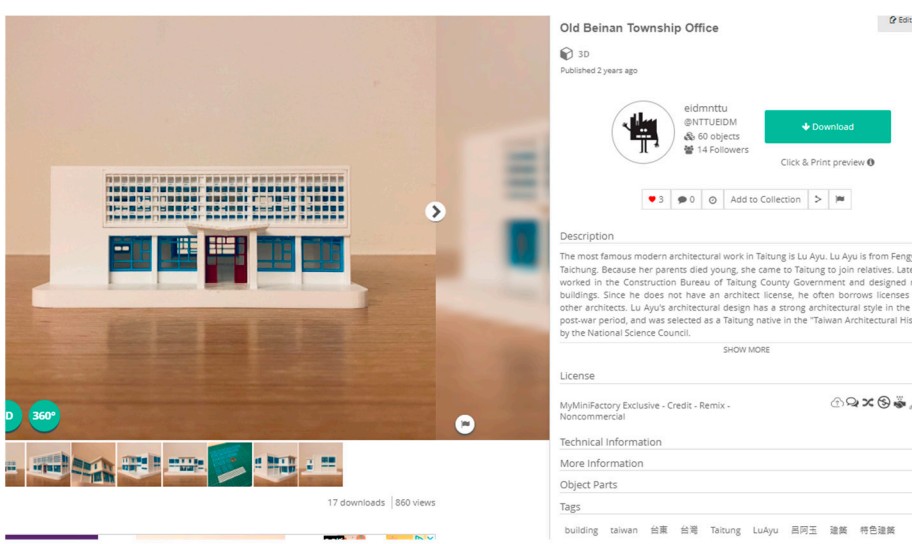

**Figure 4.** Open-source platform interface. (Source: this study).

The situation of works after uploading:

1. Since the first work was uploaded to Myminifactory (16 March 2019–10 September 2022), the total number of views reached 55,802, the number of downloads was about 1376, and the top 5 objects downloaded from this platform were: muntjac 47 times, Call of the Wild Orbiter 46 times, Deinagkistrodon 46 times, Lyudao Lighthouse 43 times, wild boars 42 times.

2. Since the first work was uploaded to the Thingiverse open-source platform (16 March 2019–10 September 2022), the total number of views reached 2581, the number of downloads was 9461, the top five objects downloaded from this platform in order were: Lyudao Lighthouse 417 times, Xuan Wu Tang in the Bombing of Master Handan 349 times, Kwong Hang Fat Information Co., Ltd., Hong Kong, China (an architectural work of A-Yu Lu) 309 times, Hibiscus sabdariffa 284 times, Cypselurus 383 times.

Each Taitung digital model uploaded to the open-source platforms includes four types: single-view digital modeling, 3D-view digital modeling, online 360-degree rotation operation, physical printing model, as shown in Figure 5; modeling and physical printing of personnel, places, and props required for the cultural festival of "the Bombing of Master Handan" were shown in Figure 6. This is to provide users with a variety of models for users to access, download, print, and use based on their needs, thus maximizing the diffusion benefit for advertising Taitung.

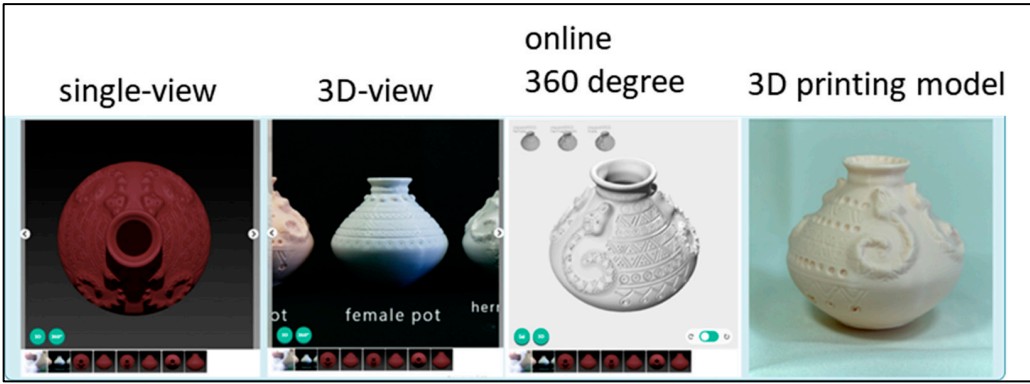

**Figure 5.** Manipulating the digital model and the printed physical model. (Source: this study).

**Figure 6.** The digital model and printed physical model of "the Bombing of Master Handan". (Source: this study).

## 4. Results and Discussion

This study took Taitung, a relatively disadvantaged area in digital resources, as an example, and proposed an operational model and a process for digitizing and open-sourcing local cultural contents for the reference of all cities and towns in Taiwan, with the expectation that all towns in Taiwan will have their own localized 3D model database. This study analyzes the application and suggestions of six junior and senior high school teachers with experience in 3D printer operation on the use of localized digital open-source models in the classroom, in the areas of indigenous culture, life technology, and visual arts.

### 4.1. The Digital Model You Would Most like to Try to Print and Reasons

Each respondent chooses the top three digital models. The six digital models the respondents were most interested in trying to print are Lyudao Lighthouse, animals (wild boars, Reeves's muntjac, Rusa unicolor), ceramic beads, ceramic pots, and Paiwan hunting daggers. They were chosen primarily because these models are well-known, have indigenous characteristics, and can be printed for display or collection. However, the respondents were not interested in 3D printing a model of plants and fish.

> I only know Lyudao Lighthouse, which is collectible and decorative (THT-C20220116).

> Taking into account the feasibility of printing in the design process, the Lyudao Lighthouse is suitable as an example for teaching "3D printed object design". The building itself is also unique and suitable for printing as a display or collection (THT-F20220119).

> Ceramic pots are very aboriginal (THB-A20220115).

> Ceramic pots can be used in classes to introduce the characteristics of the Paiwan, and the three models are very good examples (TMV-E20220119).

> The Paiwan hunting daggers are very aboriginal (THB-A20220115).

> It can be used as Paiwan specialties. Handmade hunting daggers are not easily available, so the printed ones can replace the real hunting daggers very conveniently (TMV-E20220119).

> Because of the small size of the ceramic beads, it is suitable for beginner students to learn how to operate the 3D printer and the subsequent process of removing the stand, grinding, and painting (TMT-D20220116).

> Even the details of a wild boar are there, which is very realistic! (THT-B20220116).

> These animals are so detailed and realistic that they are suitable for printing by light-curing as display materials or series collections (THT-F20220119).

### 4.2. Whether Digital Models Are Useful in Teaching or Life, and the Need for Using Digital Models

Among the five major types, the respondents thought that the "characteristic architecture" models were more helpful because the 3D models of buildings were designed in a block-like way, which made it easier for the viewers to understand the structure and characteristics of buildings, and the parts were easier to print and assemble successfully. Ceramic beads, which are in small sizes and various patterns, are also suitable for beginners

as a printing target, not only for successful printing but also for appreciating the aboriginal culture and art.

> In teaching practice, I have met many students who are interested in architecture, but currently, most of the buildings are presented in the form of flat drawings or models. It would help students differently if they could use 3D drawing and digital printing (THT-B20220116).

> Architectural models can be actually viewed in miniature, and even disassembled to make it easier for children to understand the structure and features of a building. A printed or 3D drawing is easier for children to understand than floor plans. The use of local architecture also better introduces children to local architecture (TMV-E20220119).

> From the perspective of teaching "3D modeling and printing technology" as the theme of the course, a model of a building with a "design of parts" would be a useful teaching example. In the teaching field, the author observed that when 3D modeling beginners were unable to grasp the design skills when designing 3D printed models, this would lead to failure in the 3D printing process. If I could use the work designed with the convenience of 3D printing under consideration as a solid example for the learners to observe and imitate, it could help them to improve their design skills in 3D modeling, which in turn would improve the success rate of their work when printing (THT-F20220119).

> It will help teachers of humanities courses who are not familiar with modeling to approach digital technology and allow students to appreciate aboriginal culture and art and understand the elements of creation (THB-A20220115).

> The high time cost of 3D printing and no-frills machines in schools led to a higher failure rate. However, in order to allow students to have more contact with each printing step, we must choose simple and easy-to-succeed 3D modeling. Hence, ceramic beads in small sizes were selected as they could be painted and strung into a piece of jewelry. The initial application of the course is to teach students to put the file into the modeling software, feed it into the machine, wait for the printing to be completed, and then remove the holder, polish, paint, and string it into a charm. Good responses were received from students (TMT-D20220116).

> Since the Lord Handan Festival takes place only in the city of Taitung, not all children have the experience of seeing it up close. Therefore, printing the model can help children understand the special festival in Taitung and then get a more concrete touch of it (TMV-E20220119).

In addition, some teachers suggested that the model could be more "operational" in order to explain the abstract concept of an object:

> They may want a printable model to be able to show the functioning of the objects, so the modeled objects were not usually printed (THT-C20220116).

> The 3D-printed objects should ideally illustrate abstract concepts and present objects that they cannot see or touch (THB-A20220115).

In summary, this study was conducted to build an open-source database of Taitung's cultural contents, which were divided into five categories: cultural artifacts, local ecology, characteristic plants, characteristic architecture, and cultural celebrations. They were all digitally modeled and made available to the public for free download. The models of the 60 items constructed by six different instructors have four common results: (1) High visibility does make it easy to gain favor: e.g., Lyudao Lighthouse. (2) Those models with strong local characteristics are suitable for display and collection, including Paiwan ceramic pots, Paiwan hunting daggers, and Lyudao Lighthouse. (3) For novice 3D printers, objects in small sizes and which can be combined with other objects are more suitable. A good example would be ceramic beads which are not easy to fail in printing. During

printing, students can also learn about the meaning of the indigenous ceramic bead culture. (4) Animal models come with realistic details, such as wild boars.

Based on the analysis on the respondents' areas of expertise, life science teachers, who are familiar with 3D modeling and printing capabilities, preferred models with structural components that can function and explain cultural meanings, such as Lyudao Lighthouse, over monotypic models, such as plants and fish. The teachers of aboriginal culture and visual arts with a focus on the social and humanistic aspects agreed that digital models could help students understand the local cultural content in an innovative and interesting way. However, due to the limitations of 3D modeling and printing capabilities, they paid more attention to the printing time and ease of printing of models. As a result, the small sizes of models and the designs of the parts are key factors which affect the subsequent teaching and practical applications in the field of society and humanities. This echoes Nemorin and Selwyn, 2016 [30]; Leinonen, et. al., 2000 [31], Fahrurozi et. al., 2019 [32], Schelly et. al., 2015 [33], and Song, 2019 [34].

When reviewing the connection between cultural content and digitalization, it could be found that in 1998, the National Science and Technology Council (then Ministry of Science and Technology) launched the "Digital Museum Project (1998–2002)". This project aimed to build a digital museum with local characteristics and suitable for national conditions, establish a cooperative development mechanism and environment for digital museums in Taiwan, as well as train related technical personnel and develop related industries. This event is the beginning of the digitization of cultural contents in Taiwan.

In 2002, the Ministry of Science and Technology launched the "Taiwan e-Learning and Digital Archives Program" (2002–2007), which aims to digitize valuable tangible and intangible cultural assets Academia Sinica, National Museum of Natural Science, National Palace Museum, National Museum of History, and Academia Historica. Tangible artifacts are recorded by photography, festivals and rituals are recorded by video, and songs are recorded by audio. In 2008, the Ministry of Science and Technology advanced the implementation of the Taiwan e-Learning and Digital Archives Program (2008–2012) with the expectation of building a high-quality digital learning environment. The Ministry aimed to apply the collections to the educational field and industrial applications to shorten the digital gap between urban and rural areas. Today, after more than 20 years of digitization of cultural contents, technology, and devices for 3D printing have become popular and accessible to the general public. The trends of open-source sharing and the self-maker movement are irresistible in this era.

This study not only employed the previous digital image method but also presented images in 3D models so that users can understand the culture, history, architecture, sceneries, and folklore. The 3D models can also be used as a teaching tool for history and geography teaching in Taiwan's junior high schools and elementary schools, which are very convenient and practical, while driving the economy of 3D printing related industries. In addition, this study took Taitung as a case study. Through sharing with the general public, the innovative 3D digital open-source database was used as an inversion of the rural areas, spreading Taitung's unique charm and bringing innovative industrial energy to Taitung. On one hand, the local culture can be preserved, and on the other hand, the local cultural content can be expanded into various fields of education, research, and social development. A good use of technology and emerging thinking in the local culture will promote Taitung culture internationally. Tu [4] argued that the Taiwan government, academics, and industry may refer to DAPRA/NSF grants or EU sponsorships, and that more government and academic R&D results should be open-sourced to activate the community and the overall innovation force, which can reduce development costs and form a positive ecological cycle.

To sum up, this study explores the digital open source of local cultural content, and the results are as follows:

1.　Among a large number of open-source objects, the digital model with high popularity and local distinctive features is like a signpost, which is conducive to attracting

attention to local cultural content and is suitable for users to print out for display or collection.

2.  A building model designed in parts helps viewers understand the structure and features of the building more easily, and it is easier to succeed in 3D printing and assembling. Furthermore, ceramic beads with small size and various patterns are suitable for printing beginners as printing targets. They are not only easy to print successfully, but also can appreciate aboriginal culture and art.

3.  In terms of subsequent teaching and practical applications, the printing time and ease of printing of models are still key factors in the digitization of local cultural content in terms of category selection and model design.

4.  In terms of the audience served, the cultural content, which used to emphasize professionalism and precision and focused on academic research and education by many experts, can now be extended to any person in the world, and cultural views are open and free.

5.  In terms of knowledge dissemination benefits, by breaking the boundaries of time and space, local cultural knowledge is no longer limited to local knowledge. Technology allows people who cannot visit the local area to see the Taitung cultural objects or even download and print them in real-time over the cloud platform to understand the details of the local cultural elements and their cultural connotations in a three-dimensional way. For the relatively remote Taitung, it is an innovative way to allow more people to learn the Taitung cultural contents and spread the benefits of intellectual property (IP).

6.  In terms of the significance of cultural open source, in the past, local cultural contents were passively preserved and simply enjoyed or viewed, while open source is based on culture, allowing users to actively download and apply, and allowing users the freedom to adjust the size of the object to meet the needs of their own home environment. In this way, local cultural contents are truly used for education in life, with good cultural diffusion benefits and breadth.

7.  In terms of information platforms, the previous collection platforms are all one-way browsing, where users need to go to a specific collection platform to view what they want. Nowadays, globally popular 3D model platforms are used, with complete systems, a higher number of enthusiasts, and easy-to-use interfaces. These platforms even provide such social functions as collections, comments, downloads, and sharing of users, showing higher interaction performance. Moreover, in the past, 3D modeling platforms mostly used figures, tools, and engineering parts as printing themes, but now, local cultural contents are used as digital modeling themes and to present innovative themes to the world.

## 5. Conclusions and Future Prospect

Local culture is a manifestation of people's life trajectory on the land. With the changing times, digital collections are not only digitizing traditional cultural relics but also have the mission of cultural connotation in horizontal and vertical inheritance. Digitization is seen as a "new way of organizing, structuring, and presenting knowledge". This study took Taitung as an example and proposed a model and operation process for open-sourcing local cultural contents. A total of 60 local digital models were built and displayed on two global open-source platforms, fully utilizing the function of local knowledge transfer without being restricted by geography. When resources are open and shared with the world, the chances of being noticed naturally increase. As more and more developers invest in the project, the first users can be established quickly, laying the foundation for the subsequent development of local culture digitization. With this as a model of reference, it is expected that all cities and towns in Taiwan will have their own localized 3D model database, so that the results of cultural digitization can be truly used and value-added, and local culture will no longer be just a collection of artifacts on the screen. Nowadays, 3D printers have become popular and mature in terms of hardware and technology. With

3D printers, everyone can simply print down the cultural contents in any place around the world at any time, enhancing the opportunity to know each land, enabling the public to use local culture, constructing a unique "local cultural digital experience", practicing the opportunity to make use of culture and truly breaking the gap between urban and rural areas. This study established an Object Information Cataloging Format to facilitate an effective understanding of the cultural meaning of objects and print information. It also created the first exclusive Taitung cultural content open-source database, expanding the concepts of digital manufacturing and open-source sharing from the engineering field to the cultural field. This study aims to extend the scope of local culture research and introduce new research perspectives in the field of cultural and creative industries.

The land of Taiwan is rich in diverse cultures and landscapes, and each city and town has its own cultural symbols and is known to the world with its unique geographical environment and humanities. The model proposed in this study can be applied to all cities and towns in Taiwan to collect the beautiful local culture of Taiwan through an innovative open-source platform, to spread the unique charm of remote villages through sharing, to design local industrial strategies, and to move towards cross-disciplinary knowledge integration of culture and technology. Through the open-source local digital model's characteristics of no geographical restrictions and free access, and particularly, under influence of COVID-19, even foreign people who cannot visit Taiwan in person can print local cultural contents remotely and understand Taiwan's cultural objects in a three-dimensional way, which will lead to the enhancement of local culture and the social atmosphere of local innovation and mutual benefit.

**Author Contributions:** Conceptualization, C.-L.C.; methodology, C.-L.C. and Y.S.; writing—original draft preparation, C.-L.C. and C.-L.L.; writing—review and editing, C.-L.C., C.-H.H. and Y.S. All authors have read and agreed to the published version of the manuscript.

**Funding:** This research received no external funding.

**Institutional Review Board Statement:** Not applicable.

**Informed Consent Statement:** Not applicable.

**Data Availability Statement:** Not applicable.

**Acknowledgments:** The author would like to thank all the respondents and students who provided valuable comments and suggestions and students who established digital models in this article. Thanks to the website of Taitung County Tourism Department for providing some of the figures in this article.

**Conflicts of Interest:** The authors declare no conflict of interest.

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
