# Peer review of "From Digital Collection to Open Access: A Preliminary Study on the Use of Digital Models of Local Culture"

_education, doi:10.3390/educsci13020205_

Round 1

Reviewer 1 Report

The paper describes the local cultural impact of digitisation processes, highlighting the role of open source systems. The work also presents newer 3D printing technologies and their integration implications.

The research paper is mostly descriptive and illustrative, with a few figures.

At the end of the article, the published literature is relatively limited.

The authors could use several more relevant literature sources.

Author Response

Comments

Answers or revision

1.        The paper describes the local cultural impact of digitisation processes, highlighting the role of open source systems. The work also presents newer 3D printing technologies and their integration implications.

Thanks to the reviewers for your comments, the authors have added 11 new literatures reviews, added opinions from 6 respondents, and rewritten the manuscript.

Thus, the new content has been added to the abstract as follows:

Through surveys and analyses, it could be concluded that when the models are designed in parts and are easy to print and display, it is more conducive for the models to be used in promotions and applications.

2.        The research paper is mostly descriptive and illustrative, with a few figures.

Thank you for your suggestions. The authors have added more literature in Chapters 1, 2, and 4, and an analysis of six junior and senior high school teachers’ suggestions on the application and use of 60 localized digital open source models in education has been added in Chapter 4, items 1-3, based on which three new findings were presented.

The following contents are added:

1.     Among a large number of open source objects, the digital model with high popularity and local distinctive features is like a signpost, which is conducive to attracting attention to local cultural content and is suitable for users to print out for display or collection.

2.     A building model designed in parts helps viewers understand the structure and features of the building more easily, and it is easier to succeed in 3D printing and assembling. Also, ceramic beads with small size and various patterns are suitable for printing beginners as printing targets. They are not only easy to print successfully, but also can appreciate aboriginal culture and art.

3.     In terms of subsequent teaching and practical applications, the printing time and ease of printing of models are still key factors in the digitization of local cultural content in terms of category selection and model design.

3.        At the end of the article, the published literature is relatively limited.

The authors added more literatures about open-source, digital modeling, 3D printing in Chapters “1, 2, and 4”. The following contents are added.

The following reviews and contents are added in Chapter “1. Introduction”.

The following contents are added in Chapter “1. Introduction”.

3D printing technology has been used in various fields, such as society, technology, education, and medicine, with varied performances. Tseng and Wang (2019) applied 3D printing to prosthetic limbs through teamwork and open source design, making the process faster and more relevant to users [1].  

(See line 37~39 on page 1 of the revised manuscript).

The creation of open innovation platforms by governments or citizens, as a powerful bridge to user innovation, can also be seen as a social movement [4].

(See line 70~71 on page 2 of the revised manuscript).

The following literature and contents are added in Chapter “2. Literature Review”.

In terms of practical cultural design, cultural content can be divided into three levels: physical or material, social or behavioral, and folk or religious [13].

(See line 134~135 on page 3 of the revised manuscript).

Open source is often a distributed and decentralized approach to collaborative development, and the open licensing of research and creative output is a prime example of user innovation [4].
(See line 168~170 on page 4 of the revised manuscript)

EI Bedewy, Lavicza, Haas, and Lieban (2022) provided teachers and students with the open source tool “GeoGebra” for learning the connection between math and architectural modeling, and the results of this study indicated that participants were able to solve the technical problems of model visualization more easily through open source modeling resources and 3D printing processes, thus effectively enhancing the learning opportunities in STEAM education [18].
(See line 184~189 on page 4 the revised manuscript)

In principle, digital modeling combined with 3D printing is user-centered and both are considered as active and self-directed learning methods [28]. In school teaching, Bonorden (2022) used digital models of flowers in biology classes and 3D printed three-dimensional models to solve the problem of having only 2D pictures of plants and flowers in textbooks. The three-dimensional models clearly illustrated flower structures, thus improving the limited learning environment [29]. However, the prerequisite is that this can only be done if the teacher has modeling skills [30,31].

(See line 280~287 on page 6 the revised manuscript)

Only a few professional teachers have modeling skills [32,33,34], and for most, it still takes a lot of energy and time to learn these new digital technologies [30,31].
(See line 289~290 on page 6 the revised manuscript)

Users can master what they learn. However, this can only be successfully practiced if the teacher has professional modeling skills, which is the prerequisite [30,31].

(See line 294~295 on page 6 the revised manuscript)

The following contents are added in Chapter “4 Results and Discussion”.

Tu (2002) argued that the Taiwan government, academics, and industry may refer to DAPRA/NSF grants or EU sponsorships, and that more government and academic R&D results should be open-sourced to activate the community and the overall innovation force, which can reduce development costs and form a positive ecological cycle [4].

(See line 608~612 on page 14 the revised manuscript)

4.        The authors could use several more relevant literature sources.

Authors have added more literature to explore, as described above.

Reviewer 2 Report

1. The author made 60 3D printing models of local culture in Taitung, Taiwan, and provided them for everyone to use through an open source network. His open attitude provides opportunities for everyone to use.  At the same time, the author believes that open source materials are a good way for the world to understand Taiwanese culture, which is one of the author's contributions to Taiwanese society.

2. But how is such open source material reused?  If you want to prove the effectiveness of open source, it is recommended not only to consider the openness of the scenic spot model, but also to investigate the situation after the scenic spot model information is reused, and the resulting effect.  For example, schools, governments, and communities in Taitung, using these open source materials, what have they done further?  It is suggested that the author can conduct some investigation and analysis in this regard, which will better prove the actual effectiveness of open source.

3. The paper explained the benefits of open source materials, 3D printing technology, and 60 3D printed models of Taitung’s cultural attractions (divided into two categories: tangible and intangible), but the author did not explain the purpose of the research in the article.  As a result, the thesis is closer to a work report on 3D printing.

4. The survey of Taitung culture is divided into literary and historical survey, ecological survey, landscape survey, folk art survey, and folklore survey, and generally collected important cultural attractions and cultural projects, and selected 60 3D printed ones from here.  Therefore, the investigation is mainly based on the appearance, and it is insufficient in explaining the cultural connotation in depth.  The author mentions "opening up the right to use the collections includes the collection (determining the value of the artifact), interpretation (interpreting the meaning of the artifact) and creation (using the artifact to create)", but the article is not clear about how the attractions  are interpreted.

Author Response

Comments

Answers or revision

1.        The author made 60 3D printing models of local culture in Taitung, Taiwan, and provided them for everyone to use through an open source network. His open attitude provides opportunities for everyone to use.  At the same time, the author believes that open source materials are a good way for the world to understand Taiwanese culture, which is one of the author's contributions to Taiwanese society.

Thanks for the affirmation of the reviewer. the author believes that open source materials are a good way for the world to understand Taiwanese local culture.

2.        But how is such open source material reused?  If you want to prove the effectiveness of open source, it is recommended not only to consider the openness of the scenic spot model, but also to investigate the situation after the scenic spot model information is reused, and the resulting effect.  
For example, schools, governments, and communities in Taitung, using these open source materials, what have they done further?  It is suggested that the author can conduct some investigation and analysis in this regard, which will better prove the actual effectiveness of open source.

The reviewers’ suggestion is greatly appreciated. In order to understand the use of open source models, I have added more content in Chapter 4:

This study analyzes the application and suggestions of six junior and senior high school teachers with experience in 3D printer operation on the use of localized digital open source models in the classroom, in the areas of indigenous culture, life technology, and visual arts. The detail contents are added in Section 4.1, 4.2 in revised manuscript.

(See line 470 on page12 to line 576 on page 14 the revised manuscript)

3.        The paper explained the benefits of open source materials, 3D printing technology, and 60 3D printed models of Taitung’s cultural attractions (divided into two categories: tangible and intangible), but the author did not explain the purpose of the research in the article.  As a result, the thesis is closer to a work report on 3D printing.

Thanks for the reviewer's suggestion.

The authors have proposed of the research at the end of Chapter “1. Introduction”. The following contents are added, as follows:

The purpose of this study is to build an open source database of Taitung cultural content in Taitung, which is a disadvantaged area in terms of scientific and technological resources, and to explore the print application and impact of the open source of local cultural content as a reference for the subsequent open source process of local cultural contents.

(See line 72~76 on page 2 of the revised manuscript).

4.        The survey of Taitung culture is divided into literary and historical survey, ecological survey, landscape survey, folk art survey, and folklore survey, and generally collected important cultural attractions and cultural projects, and selected 60 3D printed ones from here.  Therefore, the investigation is mainly based on the appearance, and it is insufficient in explaining the cultural connotation in depth.  The author mentions "opening up the right to use the collections includes the collection (determining the value of the artifact), interpretation (interpreting the meaning of the artifact) and creation (using the artifact to create)", but the article is not clear about how the attractions are interpreted.

Thanks for the reviewer's suggestion.

Local cultural content is divided into tangible assets and intangible assets. Tangible material is presented in concrete objects, while intangible cultural spirit and connotation often need to interpret cultural meaning through concrete objects. For example, the well-known folk event in Taitung, the bombing of Master Handan, the more firecrackers are exploded, the more prosperous it symbolizes wealth.

To avoid misunderstanding of the semantic meaning of Folk Art Survey and Folklore Survey, the authors replaced the word Folk Art Survey with “Craft Survey”.

(See Table 1 on page 3 of the revised manuscript).

“Opening up” the right to use the collections, which includes the collection (determining the value of the artifact), interpretation (interpreting the meaning of the artifact) and creation (using the artifact to create)

The above content is quoted from the Open Museum [21] of the Digital Culture Center of Academia Sinica to explain the purpose of open source cultural relics.

In order to be closer to the original intention, the author changed artifact to cultural relics.

From determining the value of cultural relics, interpreting the meaning of cultural relics, to using the cultural relics, Open Museum gradually opens up rights to users with different needs, encourages the public to participate in the open source of cultural relics, and creates a positive knowledge cycle.

Reviewer 3 Report

This is an interesting article to deepen new perspectives on local heritage. However, there needs to be more literature be cited on other similar initiatives in the international cultural context. I suggest improving the literature review. 

Among the most outstanding formal issues is the inclusion of "Source: this study."

Formulas such as "the authors" should always be included if the image is self-produced. Otherwise, the source should be attributed. Sometimes a link to the figures needs to be included. 

The authors do not include research questions. The objectives need to be clearly written, but the article's purpose is at the end of the introduction and in the methodology. I suggest making this clearer in the text, posing and answering research questions. 

Finally, the text should be reformulated to link more with the journal's vision and, in this case, to analyze the educational potential of this project. This can be glimpsed on page 14 but deserves further development.

Author Response

Comments

Answers or revision

1.        This is an interesting article to deepen new perspectives on local heritage. However, there needs to be more literature be cited on other similar initiatives in the international cultural context. I suggest improving the literature review. 

Thanks for the affirmation of the reviewer. The authors have added 11 new literatures reviews, added opinions from 6 respondents, and rewritten the manuscript.

The authors added more literatures about open-source, digital modeling, 3D printing in Chapters “1, 2, and 4”. The following contents are added.

The following reviews and contents are added in Chapter “1. Introduction”.

The following contents are added in Chapter “1. Introduction”.

3D printing technology has been used in various fields, such as society, technology, education, and medicine, with varied performances. Tseng and Wang (2019) applied 3D printing to prosthetic limbs through teamwork and open source design, making the process faster and more relevant to users [1]. 

(See line 37~39 on page 1 of the revised manuscript).

The creation of open innovation platforms by governments or citizens, as a powerful bridge to user innovation, can also be seen as a social movement [4].

(See line 70~71 on page 2 of the revised manuscript).

The following literature and contents are added in Chapter “2. Literature Review”.

In terms of practical cultural design, cultural content can be divided into three levels: physical or material, social or behavioral, and folk or religious [13].

(See line 134~135 on page 3 of the revised manuscript).

Open source is often a distributed and decentralized approach to collaborative development, and the open licensing of research and creative output is a prime example of user innovation [4].
(See line 168~170 on page 4 of the revised manuscript)

EI Bedewy, Lavicza, Haas, and Lieban (2022) provided teachers and students with the open source tool “GeoGebra” for learning the connection between math and architectural modeling, and the results of this study indicated that participants were able to solve the technical problems of model visualization more easily through open source modeling resources and 3D printing processes, thus effectively enhancing the learning opportunities in STEAM education [18].
(See line 184~189 on page 4 the revised manuscript)

In principle, digital modeling combined with 3D printing is user-centered and both are considered as active and self-directed learning methods [28]. In school teaching, Bonorden (2022) used digital models of flowers in biology classes and 3D printed three-dimensional models to solve the problem of having only 2D pictures of plants and flowers in textbooks. The three-dimensional models clearly illustrated flower structures, thus improving the limited learning environment [29]. However, the prerequisite is that this can only be done if the teacher has modeling skills [30,31].

(See line 280~287 on page 6 the revised manuscript)

Only a few professional teachers have modeling skills [32,33,34], and for most, it still takes a lot of energy and time to learn these new digital technologies [30,31].
(See line 289~290 on page 6 the revised manuscript)

Users can master what they learn. However, this can only be successfully practiced if the teacher has professional modeling skills, which is the prerequisite [30,31].

(See line 294~295 on page 6 the revised manuscript)

The following contents are added in Chapter “4 Results and Discussion”.

Tu (2002) argued that the Taiwan government, academics, and industry may refer to DAPRA/NSF grants or EU sponsorships, and that more government and academic R&D results should be open-sourced to activate the community and the overall innovation force, which can reduce development costs and form a positive ecological cycle [4].

(See line 608~612 on page 14 the revised manuscript)

2.        Among the most outstanding formal issues is the inclusion of "Source: this study."

The reviewers’ suggestion is greatly appreciated. I have added as follows in the acknowledgment:

For all pictures in this article, except for those drawn by the author, some were taken by the author on site, while others were from the website of the local public sector (Taitung County Transportation and Tourism Development Office). The author can assure that there is no copyright dispute for all pictures.

3.        Formulas such as "the authors" should always be included if the image is self-produced. Otherwise, the source should be attributed. Sometimes a link to the figures needs to be included. 

The reviewers’ suggestion is greatly appreciated.

The researcher took three photos related to the building in Figure 1, and three photos in Figure 2 related to the cultural activities are within the scope of reasonable use on the website of the Taitung County Transportation and Tourism Development Office. Links to these photos are provided bellow the photos and are also indicated in the acknowledgments. The author can assure that there is no copyright dispute for all pictures.

Note: Taitung County Transportation and Tourism Development Office website’s information opening, item 4, article iii: For the purposes of report, comment, education, research or other justifiable purposes, the user may quote the information contained in this Website and is required to expressly remark the source.

4.        The authors do not include research questions. The objectives need to be clearly written, but the article's purpose is at the end of the introduction and in the methodology. I suggest making this clearer in the text, posing and answering research questions. 

The research objectives have been rewritten more clearly at the end of Chapter 1, as follows:

The purpose of this study is to build an open source database of Taitung cultural content in Taitung, which is a disadvantaged area in terms of scientific and technological resources, and to explore the print application and impact of the open source of local cultural content as a reference for the subsequent open source process of local cultural contents.

5.        Finally, the text should be reformulated to link more with the journal's vision and, in this case, to analyze the educational potential of this project. This can be glimpsed on page 14 but deserves further development

The authors have added more literature reviews in Chapters 1, 2, and 4.

In addition, the author has added new literature, surveyed six respondents, and analyzed the results with an emphasis on educational potential in Chapter 4. The entire paper has been reorganized.

The detail contents are added in Section 4.1, 4.2 in revised manuscript.

(See line 470 on page12 to line 576 on page 14 the revised manuscript)
